# Spatiotemporal Variation of Land Surface Temperature in Henan Province of China from 2003 to 2021

**Shifeng Li [1], Zhihao Qin [1,\*], Shuhe Zhao [2], Maofang Gao [1], Shilei Li [1], Qianyu Liao [1] and Wenhui Du [1]**

1    Key Laboratory of Agricultural Remote Sensing, Ministry of Agriculture and Rural Affairs, Institute of Agricultural Resources and Regional Planning, Chinese Academy of Agricultural Sciences, Beijing 100081, China; 82101181151@caas.cn (S.L.); gaomaofang@caas.cn (M.G.); 82101182283@caas.cn (S.L.); qianyuliao@yahoo.com (Q.L.); 82101171137@caas.cn (W.D.)
2    Jiangsu Center for Collaborative Innovation in Geographical Information Resource Development and Application, School of Geography and Ocean Science, Nanjing University, Nanjing 210023, China; zhaosh@nju.edu.cn
\*    Correspondence: qinzhihao@caas.cn; Tel.: +86-135-2135-0214

**Abstract:** Land surface temperature (LST) is a key parameter closely related to various land surface processes and surface-atmosphere interactions. Analysis of spatiotemporal variation of time-series LST may provide useful information to understand eco-climatic characteristics. In this study, the spatiotemporal pattern of LST and its trend characteristics in Henan Province were examined based on MODIS LST products from 2003 to 2021. In addition, the influences of land cover types, Nighttime Light data (NTL) and Normalized Difference Moisture Index (NDMI) on LST variation were analyzed. The results indicated that: (i) The LST showed slight and rapid decreasing trend for 2004–2010 and 2018–2020, respectively, whereas an obvious increasing and slight increasing trend occurred for 2010–2013 and 2014–2018. In terms of spatial pattern, high-temperature, and sub-high-temperature were mainly distributed in the central part of the province with higher level of industrialization and urbanization at the annual, spring, summer, and daytime scales. While in fall, winter, and the nighttime, the spatial distribution of LST exhibited decreased trend from the southern part to the northern part of the province, the largest Standard Deviation (STD) was observed in summer. (ii) The interannual variation rate of LST was 0.08 °C/Y. The increasing trend mainly occurred in urban and built-up areas. At the seasonal scales, the rising rate decreased sequentially in the order of fall, winter, spring, and summer. In addition, the rising rate in the daytime was higher than that in the nighttime. (iii) LST increased along with the expansion of urban and built-up lands, except in winter. At the annual scales, 84.69% of areas with NTL data exhibited a positive correlation with LST, and NDMI in the western part with high elevation presented a significantly positive correlation to LST, while a significantly negative correlation occurred in urban and built-up areas. The cooling effect of NDMI on LST in the daytime was greater than that in the nighttime. In cropland areas, LST showed a non-significant correlation with NDMI at the annual scale, and a significantly negative correlation with NDMI in spring, summer, and fall. The influence mechanism of cropland on the variation of LST at different timescales needs to be further explored. These findings might provide some hints to understand climate change and its causes in the province.

**Keywords:** land surface temperature; spatiotemporal characteristics; different timescales; drive forces; MODIS; Henan Province



## 1. Introduction

Global warming is one of dramatic challenges currently faced by the international community [1,2]. In the context of global warming, the effects of human activities on climate change have been concerned by many scholars [3–5]. Along with the rapid urbanization and industrialization, the land surface properties are altered significantly and the regional hydrothermal environment is reshaped [6–8]. As a key parameter closely related to various

land surface processes and energy balance, LST can be an effective indicator of surface-atmosphere interactions. Therefore, it is necessary to analyze a long-term spatiotemporal variation of LST to understand the characteristics of land climate. Currently, a great deal of research focuses on spatiotemporal variation of NSAT (Near-Surface Air Temperature) or SAT (Surface Air Temperature) based on station observation. Meshram et al. (2020) examined the spatiotemporal variability of NSAT in the Chhattisgarh State of India from 1901 to 2016 [9]. Kagawa et al. (2020) explored the spatial pattern and trends of SAT in the Hawaiian Islands from 1905 to 2017 [10]. Although the station-based NSAT and SAT have the advantage of being a long time series, they cannot reflect accurately the spatiotemporal variation of temperature due to exhibiting poorly spatial heterogeneity. Compared with the traditional station-based NSAT or SAT, satellite-derived LST is a unique dataset with advantage of spatial continuity [11,12]. In addition, LST can be used as an important tracer of the change of surface characteristics because LST is more sensitive to changes of energy balance and surficial properties than that of NSAT and SAT [13,14]. Therefore, more attention should be paid to spatiotemporal variation of LST.

Moderate Resolution Imaging Spectroradiometer (MODIS) is regarded as the most popular sensor for retrieving LST due to its free available and high observation frequency [13]. With the time extension of the available MODIS LST product, there have been more and more studies produced on the spatiotemporal variation of time-series LST for different regions. Mao et al. (2017) investigated the temporal and spatial differences of LST in land, the ocean, and every continent all over the world from 2001 to 2012 [15]. NourEldeen et al. (2020) showed the spatiotemporal variation of LST for Africa from 2003 to 2017 [16]. Yu et al. (2020) presented the inter-annual spatiotemporal variation of LST for China from 2003 to 2017 [14]. Wei et al. (2021) analyzed the LST variation in the agricultural pastoral ecotone of northern China from 2003 to 2020 [13]. Xing et al. (2020) examined the year-to-year variations of LST under clear-sky conditions for the whole world based on annual temperature cycle (ATC) model [17]. From the above studies, the LST presented an increasing trend inter-annually. Generally, the variation of LST in different seasons and times of day and night generate different performance [18], so the diversity of spatiotemporal variation of LST need to be concerned. In addition, different regions have different responses to climate change. Therefore, it is necessary to investigate on this topic further for different regions at different timescales.

The driving forces of LST variation have attracted increasing attention. In previous studies, researchers mainly focus on the correlation between LST and LUCC (Land Use/Cover Change) and NDVI (Normalized Difference Vegetation Index) [19–21]. The influences of moisture and socio-economic development on time-series LST variation were rarely investigated based on the remote sensing dataset. NDMI is an effective indicator of land surface moisture and has a closer relation to LST than NDVI, which was reported in several studies [22,23]. The socio-economic development means population agglomeration and energy consumption increase to some extent, which cause the increase in carbon emission and influence LST variation. Nighttime Light (NTL) data can reflect effectively socio-economic development [24] and it was widely used in population and economic development estimation [25,26], urban, and spatial structure [27].

Henan Province of China has experienced rapid urbanization since the strategy of "the Rise of Central China" was put forward. The urban and built-up areas of the province was 1345 km$^2$ in 2003 and it increased to 3040 km$^2$ in 2020 [28]. Such changes of surface characteristics had an obvious impact on LST variation in the province. What will happen to LST at different timescales along with the change of surface characteristics in Henan Province? In addition, Henan Province is also a major agricultural province in China; the growth of crops is closely related to temperature. Therefore, analyzing the spatiotemporal variation of long-time series LST in Henan Province of China can provide an important demonstration of regional change and a reference for agriculture production.

The primary objectives of the study are to (i) analyze the spatiotemporal variation of LST at different timescales in Henan Province from 2003 to 2021, and (ii) investigate the

drive mechanism of LUCC, NTL, and NDMI on LST variation. The conceptualized diagram of the study, as shown in Figure 1, and the structure presents as follows: information on the study region, data resources, and methods will be presented in the Section 2, which will be followed by the results and discussion in Section 3. Finally, we end with a conclusion section to highlight the key points of the study.

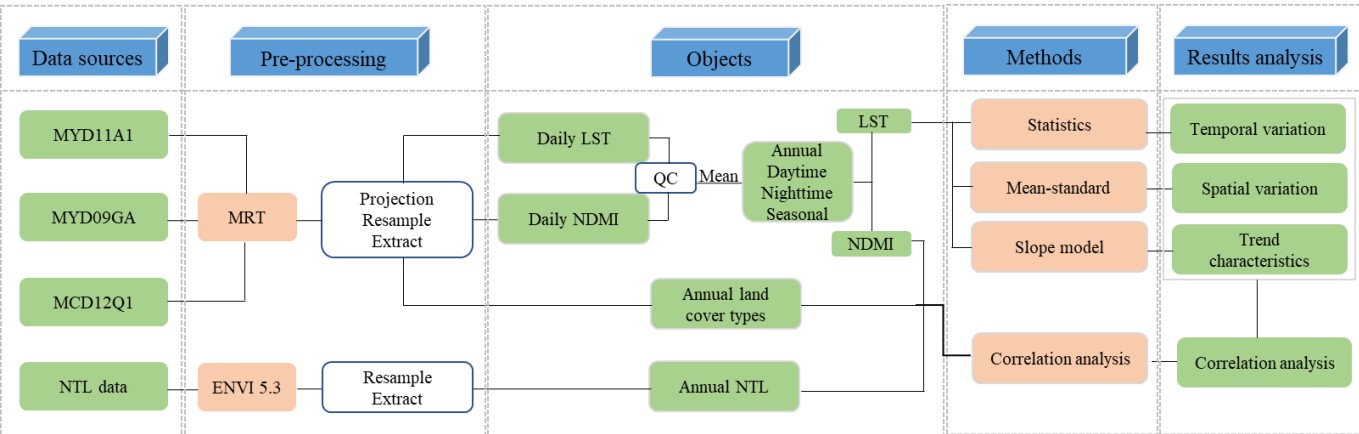

**Figure 1.** The conceptualized diagram of the study.

## 2. Materials and Methods

### 2.1. Study Region

Henan Province, located in the middle of China, ranges from 110°21′ E–116°39′ E, 31°23′ N to 36°22′ N, with a total area of $16.7 \times 10^4$ km$^2$. The elevation ranges from −135 m to 2387 m; terrain presents high in the west and low in the east (Figure 2). The western part of the province is dominated by mountainous and hilly terrain, while the southwestern part and central-eastern part in the province are dominated, respectively, by Nanyang basin and the Huang-Huai-Hai alluvial plain. Henan Province has a temperate continental monsoon climate, with four distinct seasons, rain and heat over the same period. It is windy and dry in spring, hot and rainy in summer, cool and sunny in autumn, and cold and less rainy in winter. The annual average air temperature ranges from 12.9 °C to 16.5 °C and the highest temperature occurs in July. The annual average rainfall varies from 464.2 mm to 1193.2 mm and it is mainly concentrated from June to August [29].

Henan Province is a major agricultural province in China; land cover is dominated by croplands (Figure 3). The annual average yield of the grain accounts for 66.49 million tons, and accounts for nearly 1/10 of the whole China [30]. Henan Province is also a populous province in China, with a resident population of 99.36 million and a population density of 595 people/km$^2$ [28]. In addition, Henan Province governs 18 prefecture-level cities, including Zhengzhou city, Luoyang city, and Kaifeng city, etc., and most of urban and built-up lands are mainly distributed along the Beijing-Guangzhou railway, Lianyungang-Lanzhou railway, and Jiaozuo-Liuzhou railway lines (Figure 3).

### 2.2. Data Sources and Pre-Processing

#### 2.2.1. MODIS Data

In this study, MODIS products (Version 006) including MYD11A1, MYD09GA and MCD12Q1 covered study region were collected from the website https://ladsweb.modaps.eosdis.nasa.gov/ (accessed on 1 March 2022).

MYD11A1 was used to examine the LST variation in Henan Province at different timescales during the period from 2003 to 2021. It was reported that the MYD11A1 LST product was retrieved using the split-window algorithm and its error was less than 1 K [31]. Several studies proved that the change of extreme value of LST had more obvious influence on drought, ice melting and phenology [32–34]. Therefore, MYD11A1 was selected in this

study due to the daytime LST and the nighttime LST contained in MYD11A1 are closer to maximum value and minimum value of daily LST [35–37].

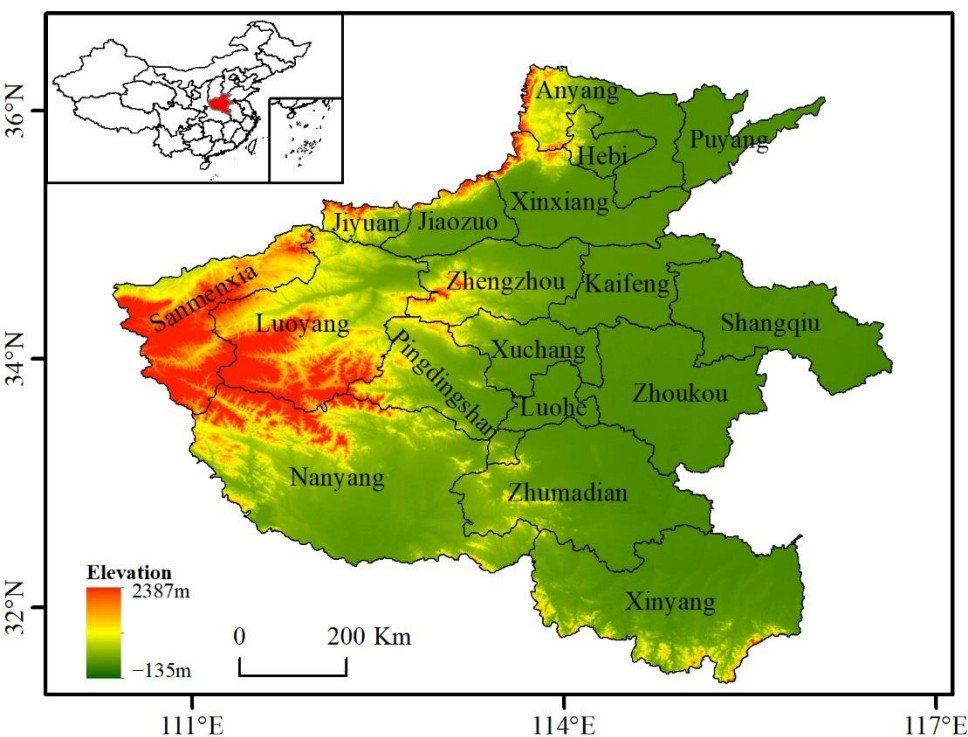

**Figure 2.** DEM and administrative division of Henan Province, with the inserted map showing the geographical location of the province in China.

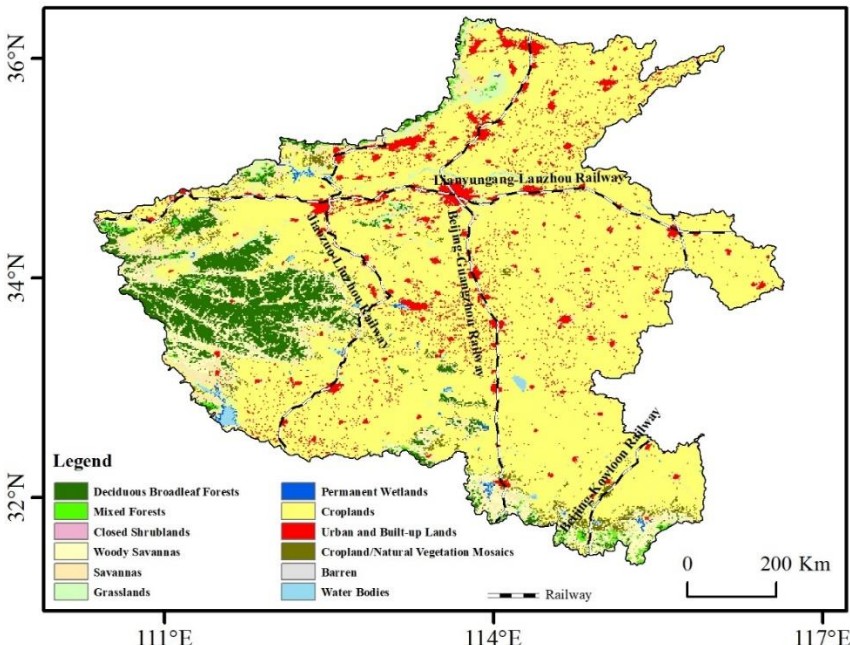

**Figure 3.** Land cover types and principal railway lines in Henan Province, China.

The LSR of the mid-infrared band and the red band at 500 m resolution contained in MYD09GA during the period from 2003 to 2021 were selected to calculate NDMI (Normalized Difference Moisture Index), and assess the effect of NDMI on LST variation in this study. Generally, the value is closely related to water supply sufficiency [38,39]. Figure 4

shows the multi-year average NDMI value for 2003–2021 in Henan Province. As shown in Figure 4, high values are mainly distributed in water bodies and croplands located in central and eastern part of the province, while a low value is mainly distributed in urban and built-up lands along the railway lines and the western part with high elevation.

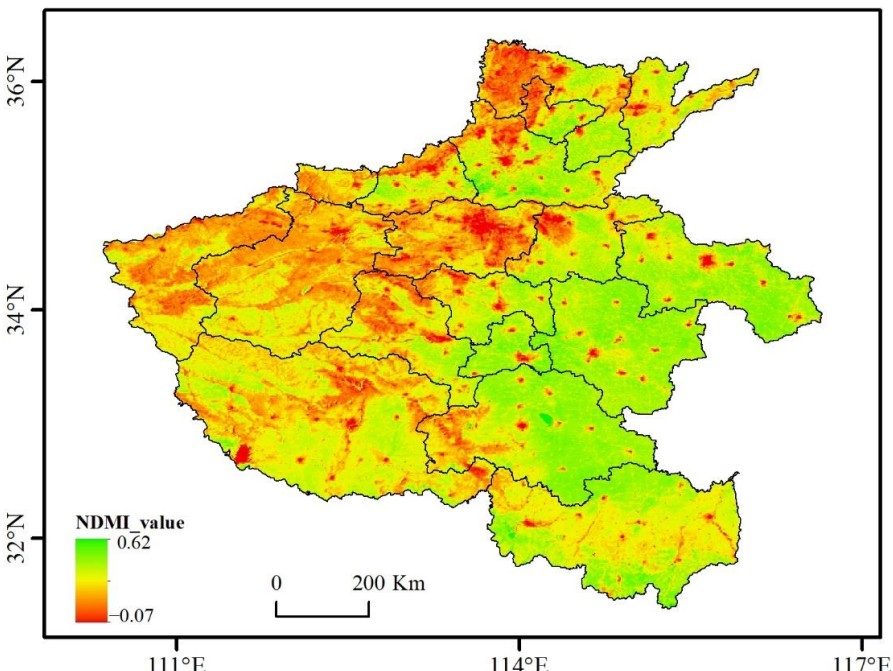

**Figure 4.** Multi-year average NDMI value for 2003−2021 in Henan Province, China.

The MCD12Q1-IGBP (The International Geosphere-Biosphere Program) land cover type product at 500 m spatial resolution during the period from 2003 to 2020, considering data accessibility, was chosen to analyze the LST characteristics among different land types. The IGBP product provides 17 classes of land types at an annual time step for the whole world. As shown in Figure 3, there are 12 classes land types in Henan Province. However, some land types, such as closed shrublands and wetlands have few pixels and lack representativeness. Therefore, the 12 classes land types were grouped into seven classes according to the similarity. The classes of deciduous broadleaf forests, mixed forests, and closed shrublands were categorized as forestlands. Woody savannas and savannas were classified as woodlands. Permanent wetlands and water bodies were grouped as wetlands, and croplands and cropland/natural vegetation mosaics were categorized as croplands.

The pre-processing for MODIS data included three steps: (i) Projection, resample, and format transformation for MODIS products were carried out by MRT (MODIS Reprojection Tool), and the coordinate system of the image was unified as WGS-84 benchmark geographic coordinates, spatial resolution was unified as $0.01° \times 0.01°$; data format was unified as TIF. (ii) In order to ensure the accuracy of analysis, LST and LSR products were controlled by QA (Quality Assurance) and the pixels with good quality were remained only by IDL 8.5 tool (IDL, Interactive Data Language Version developed by Harris Geospatial Solutions, Inc., Boulder, CO, USA). (iii) The DN (Digital Number) of pixels with good quality were transformed into the actual value of LST expressed by Celsius degree (Equation (1)) and actual LSR (Equation (2)), respectively, and daily NDMI was calculated (Equation (3)). Finally, the LST and NDMI at different timescales were obtained by the average method [17]. Each season had three months, spring (March, April, and May), summer (June, July, and August), autumn (September, October, and November), and Winter (December, January, and February).

$$LST = DN \times 0.02 - 273.15 \tag{1}$$

$$LSR = DN/10000.0 \tag{2}$$

$$NDMI = (NIR - MIR)/(NIR + MIR) \tag{3}$$

### 2.2.2. Nighttime Light Data

In this study, NPP-VIIRS-like NTL data were used to explore the influence of socio-economic activities on LST variation. NPP-VIIRS-like NTL data were developed from cross-sensor calibration by Chen et al. in 2020 [24]. Nowadays, it provides NTL data at annual time step and 500 m spatial resolution from 2000 to 2020, and can be downloaded freely at the website https://doi.org/10.7910/DVN/YGIVCD (accessed on 10 June 2021). The data format is TIF; the coordinate system is WGS-84 benchmark geographic coordinates. Considering the accessibility of NPP-VIIRS-like NTL data, we only analyze the relation to LST from 2000 to 2020 at annual scale. Before the analysis, the spatial resolution was resampled to $0.01^{\circ} \times 0.01^{\circ}$ to keep consistent with MODIS LST. Figure 5 presents the multi-year average NTL brightness value for 2003−2020 in Henan Province. As shown in Figure 5, a high brightness value is basically consistent with urban and built-up lands with high level urbanization along the railway lines.

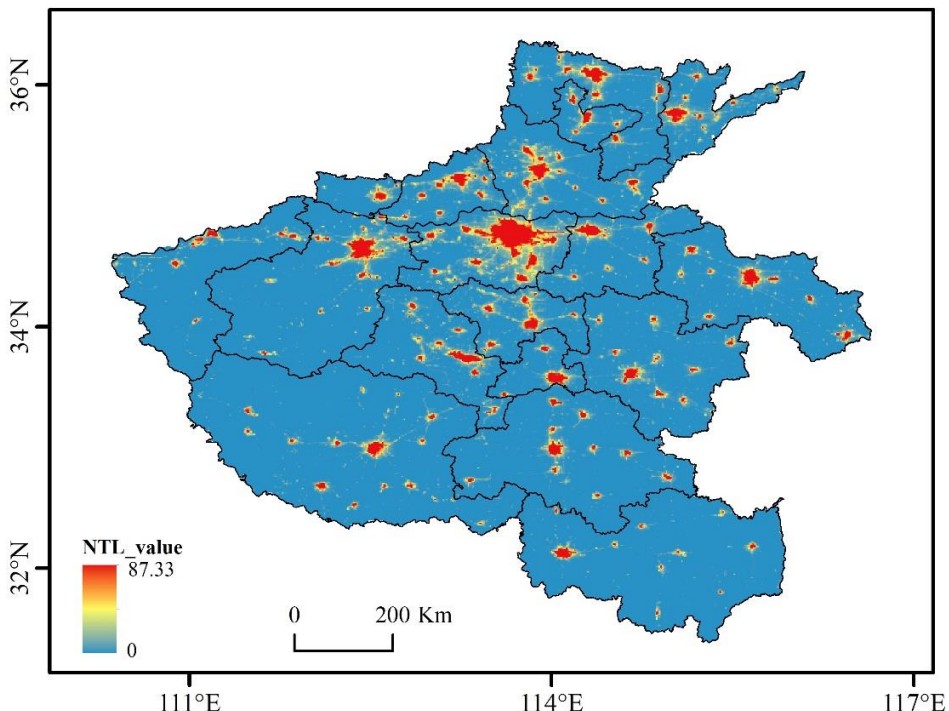

**Figure 5.** Multi-year average NTL brightness value for 2003−2020 in Henan Province, China.

### 2.3. Methods

#### 2.3.1. Classification of LST

The LST was divided into five grades by the mean-standard deviation method [40] to analyze the spatial variation for Henan Province, from high to low, followed by high temperature, sub-high temperature, medium temperature, sub-low temperature, and low temperature (Table 1).

**Table 1.** The classification criteria of LST.

| Grades of LST | Value Range |
| --- | --- |
| High temperature | $T_S \geq \mu + 1.5 \times S$ |
| Sub-high temperature | $\mu + 0.5 \times S \leq T_S < \mu + 1.5 \times S$ |
| Medium temperature | $\mu - 0.5 \times S \leq T_S < \mu + 0.5 \times S$ |
| Sub-low temperature | $\mu - 1.5 \times S \leq T_S < \mu - 0.5 \times S$ |
| Low temperature | $T_S < \mu - 1.5 \times S$ |

$T_S$ is the LST value of the pixel; $\mu$ is the average LST; $S$ is standard deviation (STD).

### 2.3.2. Spatiotemporal Variation of LST

The slope analysis method was used in this study to investigate the variation trend of LST in the province [13,16]. The slope expression is as follows:

$$Slope = \frac{19 \times \sum\limits_{i=1}^{19} (i \times T_S) - \sum\limits_{i=1}^{19} i \times \sum\limits_{i=1}^{19} T_S}{19 \times \sum\limits_{i=1}^{19} i^2 - \left(\sum\limits_{i=1}^{19} i\right)^2} \tag{4}$$

where $T_S$ is the LST value of the pixel, $i$ is the time series from 2003 to 2021 in this study, and the value is taken as $1-19$. Generally, $Slope > 0$ indicates that the LST shows a upward trend, and $Slope < 0$ indicates that the LST shows a downward trend.

F-test was used in this study to evaluate the significance level of the change trend [41]. The expression of F-test is as follows:

$$F = U \times \frac{n-2}{Q} \tag{5}$$

$$U = \sum_{i=1}^{19} (\hat{T}_{SI} - \overline{T}_S)^2 \tag{6}$$

$$Q = \sum_{i=1}^{19} (T_{si} - \overline{T}_s)^2 \tag{7}$$

where $U$ represents the regression square sum, $Q$ represents the residuals of the regression square sum, and $n$ represents the total number of years, which is taken as 19 in this study, $\hat{T}_{SI}$ represents the regression value of LST in $i$th year, $\overline{T}_S$ represents the average LST from 2003 to 2021, and $T_{si}$ represents the value of LST in $i$th year.

Based on slope and F-test, the variation trend of LST in Henan Province is divided into five grades: significant increase ($Slope > 0$, $p \leq 0.01$), slight increase ($Slope > 0$, $p \leq 0.05$), non-significant variation ($p > 0.05$), slight decrease ($Slope < 0$, $p \leq 0.05$), and significant decrease ($Slope < 0$, $p \leq 0.01$).

### 2.3.3. Correlation Analysis

Pearson correlation coefficient was used in this study to analyze correlation between LST and drive forces including NDMI and the NTL variable [14,17]. The calculation equation is as follows:

$$R = \frac{n \times (x_i - \overline{x}) \times (y_i - \overline{y})}{\sqrt{\sum\limits_{i=1}^{n} (x_i - \overline{x})^2} \times \sqrt{\sum\limits_{i=1}^{n} (y_i - \overline{y})^2}} \tag{8}$$

where $R$ is the Pearson correlation coefficient defined as the correlation between two variables. Value of the coefficient ranges from $-1$ to 1. When $R > 0$, the two variables have a positive correlation, and $R < 0$, negative correlation existed between the two variables. In addition, the closer the absolute value of R is to 1, the greater the correlation between the two variables. $n$ represents the number of samples; $i$ is the time series, and the value is taken as $1 - n$.

T-test was used in this study to test the significance of the correlation between two variables. The expression of T-test is as follows:

$$T_C = \frac{R \times \sqrt{n-2}}{\sqrt{1 - R^2}} \tag{9}$$

The statistic $T_C$ follows the distribution with the degree of freedom $(n - 2)$ and gives the significance level $\alpha$, if $|T_C| > T_n$, the original hypothesis was rejected and two variables

were regarded significant correlation; otherwise, they presented non-significant correlation. In this study, all significant correlation were tested at the 0.05 of significance level.

## 3. Results and Discussion

### 3.1. Temporal Variation of LST

The change of LST at different timescales exhibited obvious fluctuation trend (Figure 6). The inter-annual change of average LST varied from 15.30 °C in 2010 to 18.96 °C in 2018, and the multi-year average value was 16.70 °C for 2003−2021 in the study region. The LST showed decreasing trend slightly from 2004 to 2010, after then, an obvious increasing trend was found for 2010−2013 and a slight increasing trend occurred for 2014−2018, whereas the LST exhibited an obvious decreasing trend from 2018 to 2020 (Figure 6a), which was closely related to the internal variability of meteorological factors and external forces produced by the Pacific decadal oscillation [13,42,43].

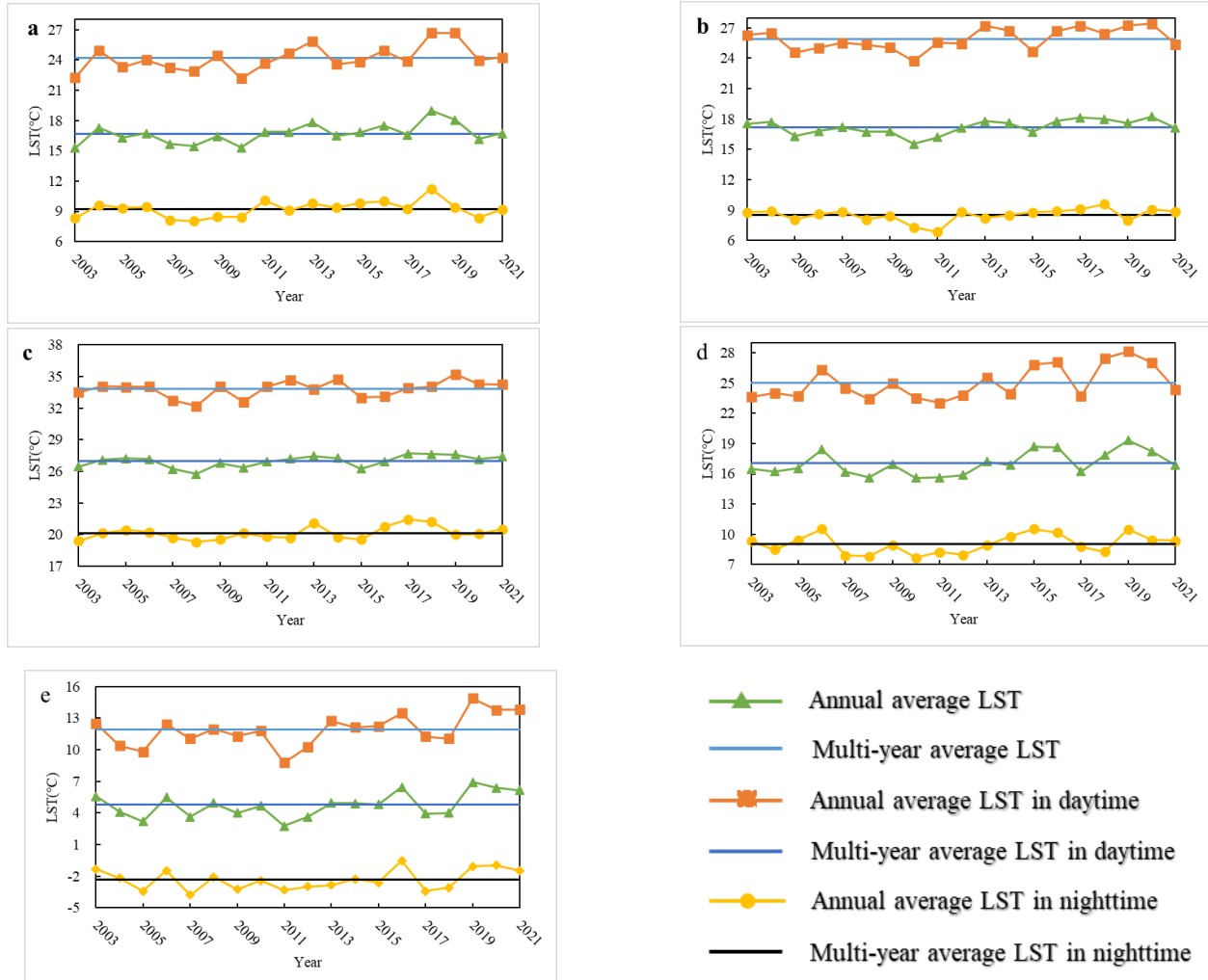

**Figure 6.** Temporal variation characteristics at different scales: (**a**) annual; (**b**) spring; (**c**) summer; (**d**) fall; and (**e**) winter.

The variation of LST between the seasons showed an obvious difference (Figure 6b–e), which was caused by climatic features of different seasons. The multi-year average LST increased sequentially followed the order of winter, fall, spring, and summer. The maximum fluctuation amplitude occurred in winter, with a value of 4.16 °C, while the minimum fluctuation amplitude occurred in summer with a value of 1.94 °C, due to high temperature widespread in summer for study region.

There were obvious differences in the variation of LST between the daytime and the nighttime. The LST difference between the daytime and the nighttime reached their maximum and minimum in spring and summer, with a value 17.40 °C and 13.66 °C, respectively. The change trend of LST both in the daytime and the nighttime was basically consistent with the annual change (Figure 3). The fluctuation ranges of LST in the daytime and the nighttime were 4.50 °C and 3.21 °C, respectively. It indicated that the variation of LST in the daytime was greater than that in the nighttime, which lies in that it is easy to be disturbed by solar radiation during the daytime [35].

### 3.2. Spatial Variation of Average LST

The multi-year average LST ranged from 6.33 °C to 20.57 °C for 2003−2021. It was found that the area proportion of medium-temperature reached to 46.50%, it was mainly distributed in eastern part of the province, including such cities as Zhoukou, Shangqiu, and Zhumadian. The land cover was dominated by croplands. Followed by high-temperature area and sub-high-temperature area, the total proportion was 29.06%, which mainly distributed in central part of the province, including such cities as Zhengzhou, Luoyang, and Pingdingshan with a higher level of industrialization and urbanization. In addition, the Nanyang basin was also dominated by a high-temperature area and sub-high-temperature area, which lies in the terrain that is high around the middle and low in the middle; the atmosphere is thick, so the heat is not easy to dissipate [44]. The low-temperature area and sub-low-temperature area mainly distributed in western part of the province due to the influence of elevation [45], including the region at the junction of Sanmenxia city, Luoyang city, and Nanyang city, as well as the region along the provincial boundary of northwestern parts (Figure 7).

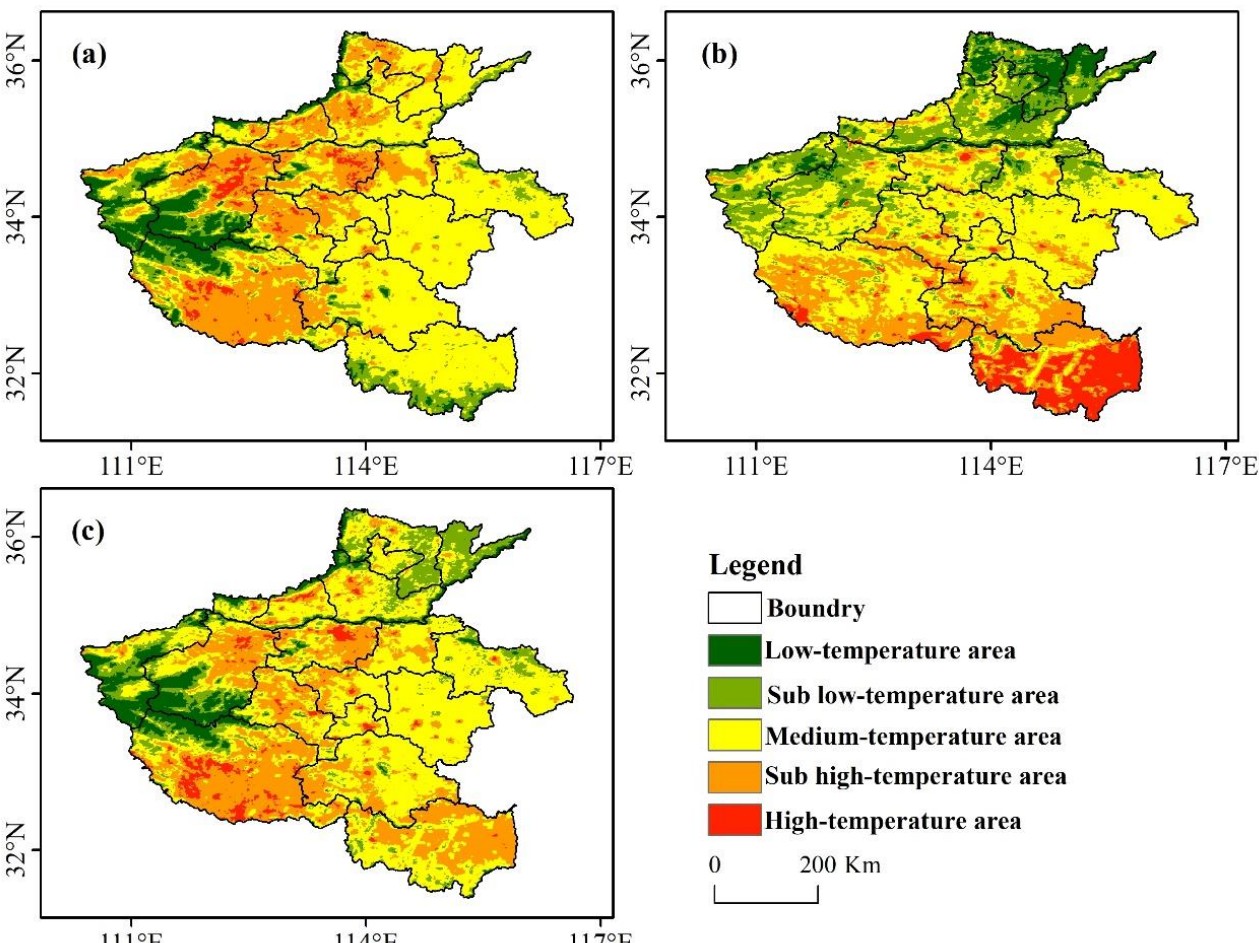

**Figure 7.** Spatial distribution of LST for 2003−2021: (**a**) daytime; (**b**) nighttime; (**c**) annual average.

The average LST in the daytime was between 9.9 °C and 31.11 °C, and its spatial distribution was similar to that of multi-year average LST (Figure 7a), while the average LST in the nighttime changed from 2.63 °C to 16.19 °C, and its spatial distribution exhibited decreased from southern part to northern part of the province on the whole (Figure 7b). It is consistent with the decrease of solar radiation energy from south to north. In addition, the high-temperature area and sub-high-temperature area decreased significantly in the central part of the province, which perhaps lies in the thermal properties of urban and built-up lands with heating quickly in the daytime while cooling fast in the nighttime [46].

The spatial distribution in spring and summer were similar to that in the daytime, but the eastern part of the province was dominated by sub-low-temperature area in spring (Figure 8a,b). While the spatial distribution in fall and winter were similar to that in the nighttime, the low-temperature area and sub-low-temperature area increased obviously in the northeastern part of the province in winter (Figure 8c,d). In addition, the highest St. DEV. (Stand Deviation) occurred in summer, which means the highest spatial variation of the province in summer. One interesting phenomenon was that, although more solar radiation energy was received in the summer, the total proportion of the high-temperature area and sub-high-temperature area was not the highest; the reason was that there was also a lot of precipitation in summer, and precipitation exhibited a negative correlation to LST [47].

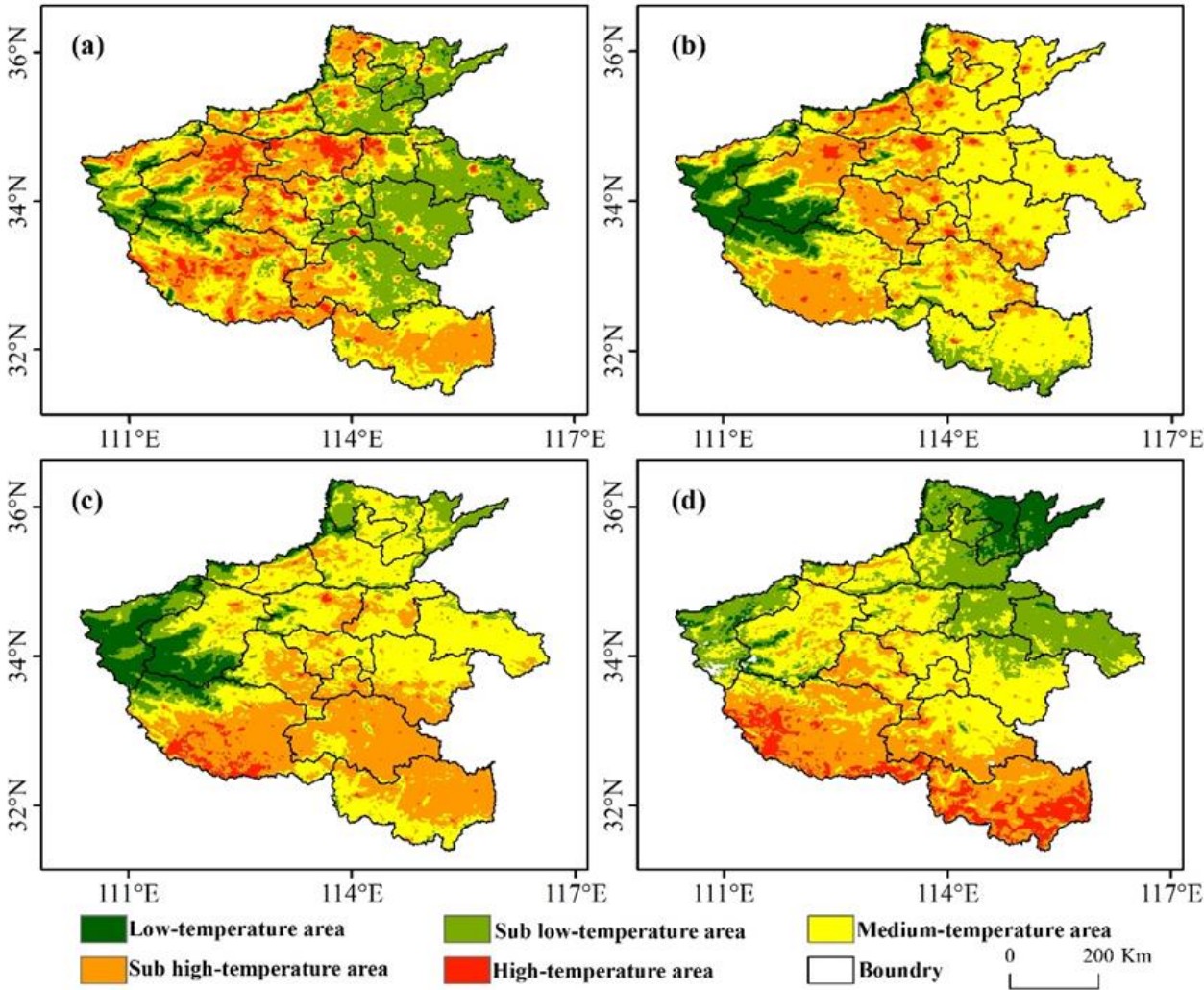

**Figure 8.** Characteristic of LST variation at seasonal scale (**a**) spring; (**b**) summer; (**c**) autumn; and (**d**) winter.

### 3.3. Trend Variation Characteristics of LST

The interannual variation rate of LST ranged from $-0.09$ °C/Y to 0.30 °C/Y for 2003−2021 in the study region. The average value was 0.08 °C/Y, and it showed an increasing trend on the whole. The rising rate was higher than that in northern China [13]. About 99.04% of the region exhibited an increasing trend (Slope > 0), with 0.99% and 24.88% of region increasing significantly ($p \leq 0.01$) and slightly ($p \leq 0.05$), respectively. They were mainly distributed in built-up areas located in such cities as Zhengzhou, Xuchang, and Luoyang Kaifeng along railway lines (Figure 9a). The LST exhibited an upward trend in both the daytime and the nighttime, and the increasing rates were 0.11 °C and 0.05 °C, respectively. In the daytime, the LST showed a significant increase and a slight increase in 2.49% and 30.79% of the areas; the spatial distribution expanded continuously (Figure 9b). Meanwhile, in the nighttime, the proportion shrunk sharply, and a significant increase mainly occurred in water bodies located in the southwestern part of the province (Figure 9c); this means that the cooling effect of water bodies was gradually weakened, perhaps in the nighttime.

At seasonal scales, the warming rate in fall and winter was higher than that in spring and summer, with values were 0.10 °C/Y and 0.09 °C/Y, respectively. This was basically consistent with the predictions of previous studies [13,48]. The lowest warming rate was in summer, with a value of 0.047 °C/Y. It was perhaps because of that precipitation was mainly concentrated in summer, which slowed down the rise of LST to a certain extent. In addition, the areas with significant and slight increase in fall and winter were distinctly less than that in spring and summer (Figure 9d–g). It meant that the more solar radiation energy received, the more obvious the heating effect for the urban and built-up lands.

The change rate of the daytime and the nighttime were further computed at seasonal timescales (Table 2). The LST in the daytime and the nighttime for four seasons showed upward trend on the whole. However, the rising rate in the daytime was higher than that in the nighttime, and the maximum difference in change rate between the daytime and the nighttime occurred in fall and winter, with the values being 0.10 °C/Y and 0.09 °C/Y, respectively, which indicated that the LST difference between the daytime and the nighttime was gradually increasing.

**Table 2.** LST variation of daytime and nighttime at seasonal scale.

|  | Range (°C/Y) | | Mean (°C/Y) | |
| --- | --- | --- | --- | --- |
|  | **Daytime** | **Nighttime** | **Daytime** | **Nighttime** |
| Spring | $-0.70-0.71$ | $-0.12-0.45$ | 0.08 | 0.03 |
| Summer | $-0.42-0.48$ | $-0.08-0.39$ | 0.05 | 0.05 |
| Autumn | $-0.20-0.58$ | $-0.18-0.53$ | 0.15 | 0.05 |
| Winter | $-0.16-0.45$ | $-0.16-0.59$ | 0.14 | 0.05 |

### 3.4. Effect of Different Land Types on LST Variation

In order to assess the effect of land cover on LST, the average LSTs of different land cover types were calculated (Table 3). The LST differences among land cover types were similar in the annual, spring, summer, and fall groups, but different from winter. In the annual, spring, summer, and fall groups, the highest LST was observed in urban and built-up land, while the highest LST occurred in wetland in winter (Table 3). The maximum LST difference among land cover types occurred in summer; the value reached 5.63 °C, which explained why highest STD occurred in summer. The average LST and STD of different land covers in the daytime and the nighttime at seasonal scales were also calculated (Figure 10). The highest LST was observed in urban and built-up land in the daytime while, that occurred in wetland in the nighttime due to the physical characteristics. In addition, we could clearly see that the STD of each land cover type in the daytime was higher than that in the nighttime; this meant the spatial variation of LST in the daytime was more obvious than that in the nighttime.

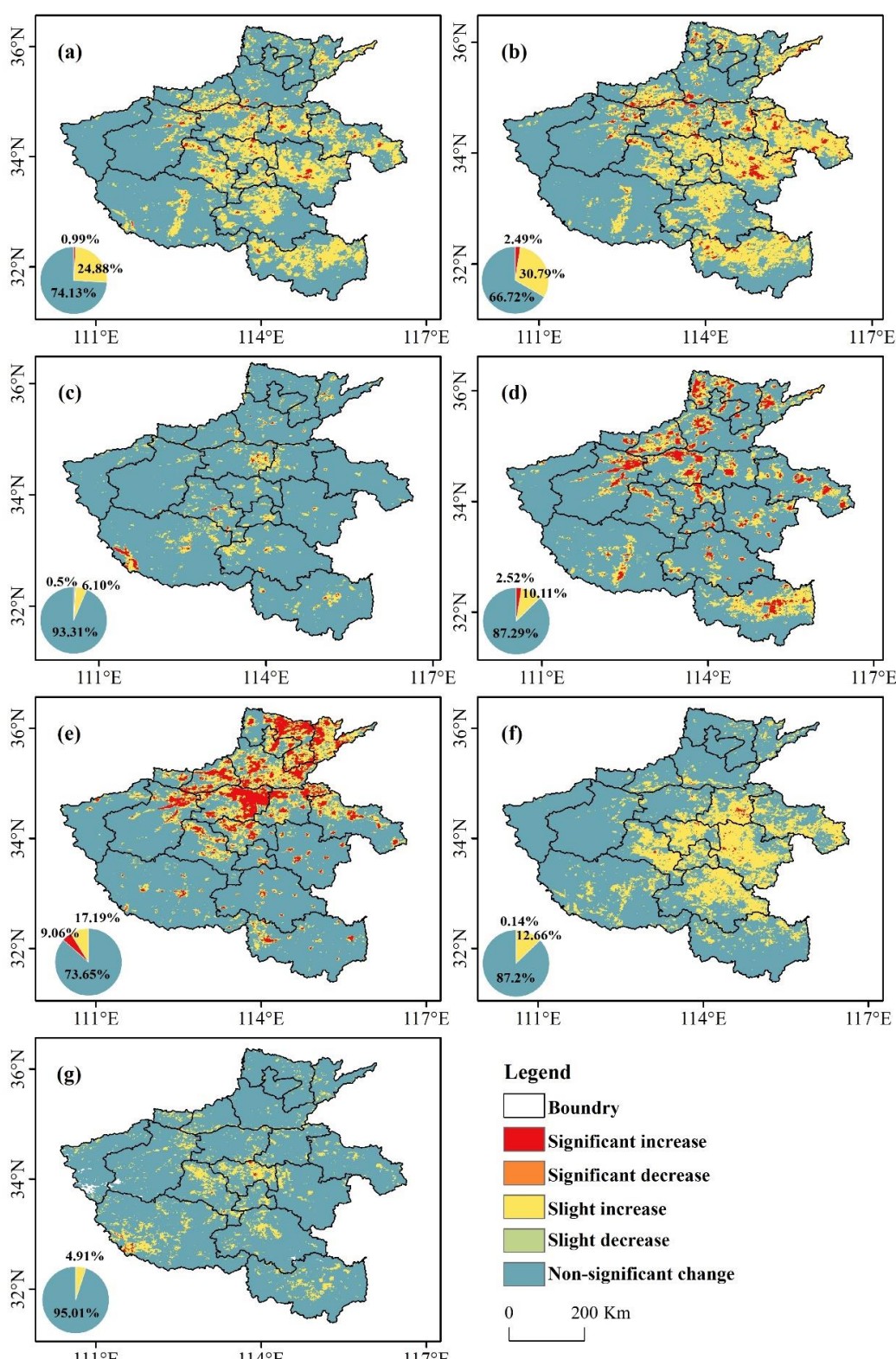

**Figure 9.** Trend variation characteristic of LST for 2003–2021: (**a**) interannual; (**b**) daytime; (**c**) nighttime; (**d**) spring (**e**) summer; (**f**) autumn; (**g**) winter.

**Table 3.** Average LST of different land types at different timescales (°C).

|  | Annual | Spring | Summer | Fall | Winter |
|---|---|---|---|---|---|
| Forestland | 14.23 | 15.65 | 23.26 | 14.14 | 4.32 |
| Woodland | 16.21 | 17.46 | 25.83 | 16.20 | 5.45 |
| Grassland | 15.99 | 17.40 | 26.29 | 16.01 | 4.41 |
| Wetland | 16.16 | 16.31 | 26.43 | 17.85 | 5.91 |
| Cropland | 16.89 | 17.63 | 26.95 | 17.14 | 5.21 |
| Urban and built-up land | 17.72 | 18.53 | 28.89 | 17.90 | 4.78 |
| Barren | 15.77 | 16.40 | 26.96 | 17.16 | 4.19 |

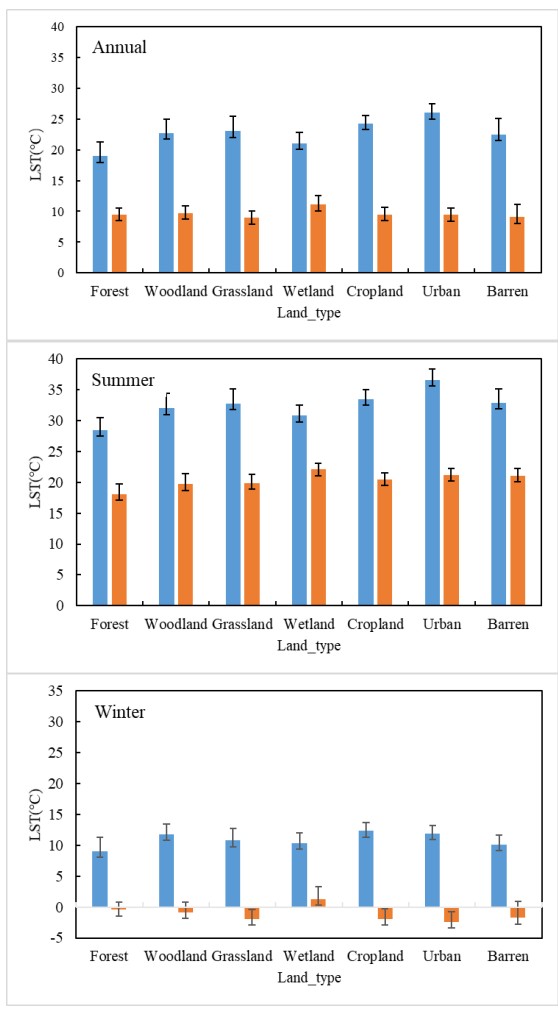

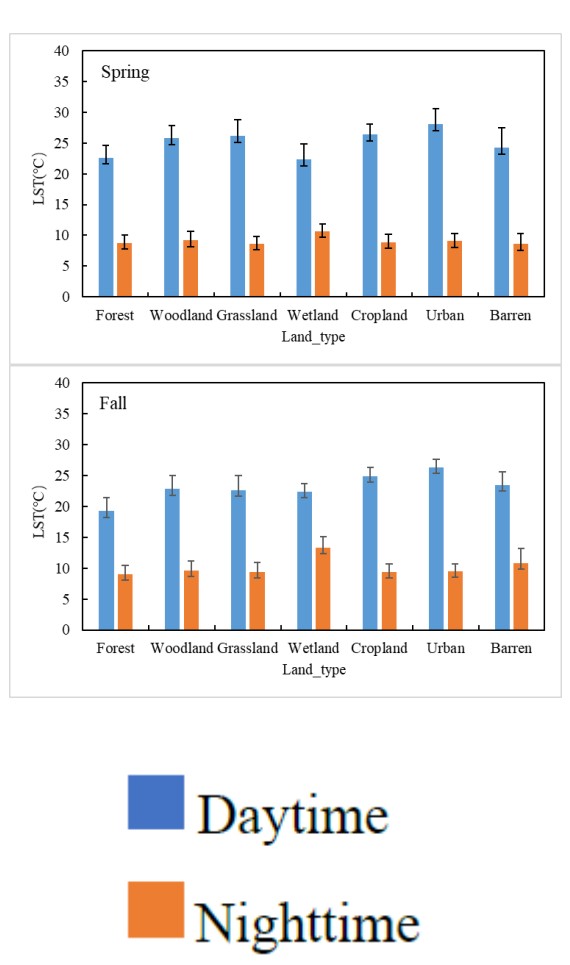

**Figure 10.** LST characteristics of different land types between daytime and nighttime.

The effect of urban and built-up land on LST at different time scales were further analyzed. As mentioned above, the LST of urban and built-up land was higher than any other land cover types in the annual, spring, summer and fall groups; this meant that the expansion of urban and built-up lands will lead to the increase of LST. In order to prove it, the pixel numbers variation of urban and built-up lands in areas with LST increase at different timescales were calculated. As shown in Figure 11, the pixel numbers in the urban and built-up lands increased significantly from 2003 to 2020 in the areas with LST increasing, especially in summer. Therefore, the conclusion was made that the increase in urban and built-up land could have caused the LST increase, which is similar to that of previous studies [6–8]. However, it is not applicable in winter, because the LST of urban and built-up lands was not the highest in winter (Table 3); in addition, the pixel numbers

of urban and built-up lands in areas with LST increased, and kept steady in winter from 2003 to 2020 (Figure 11), so the increase of urban and built-up land could not lead to the LST increase.

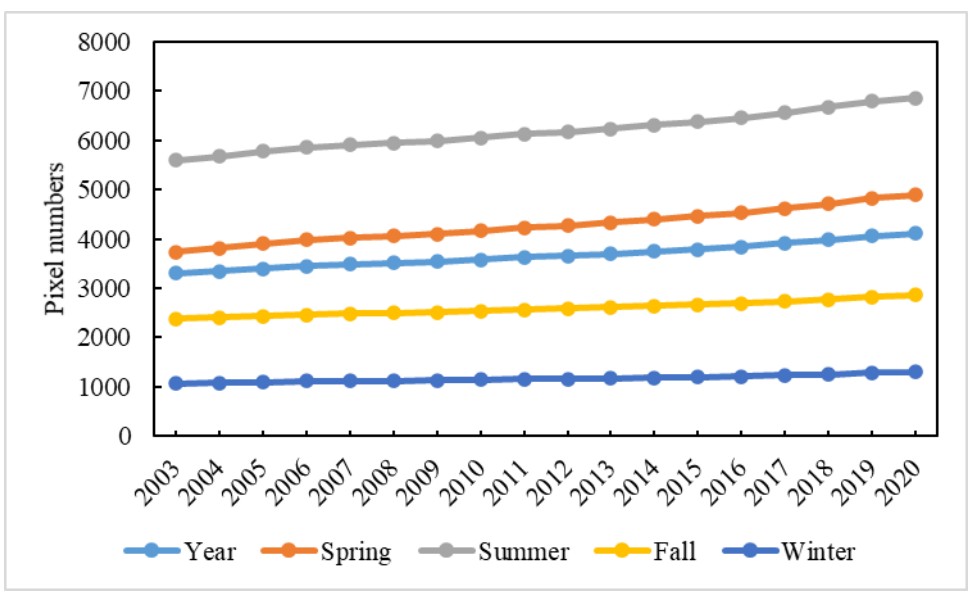

**Figure 11.** The pixel numbers variation of urban and built-up lands in areas with LST increase at different time scales.

### 3.5. Effect of Socio-Economic Factors on LST Variation

The relation coefficient (R_coefficient) between NTL and LST ranged from $-0.69$ to $0.88$ for 2003$-$2020 (Figure 12a). About 15.31% of the region showed a negative relation (R_coefficient $< 0$), mainly distributed in the urban areas with the high vegetation cover and suburbs (Figure 12a). However, the negative relation did not pass the test of significance ($p < 0.05$), indicated that the negative relation was not obvious. While the positive relation (R_coefficient $> 0$) accounts for 84.69%, which meant that the greater the brightness value of NTL in these areas, the higher the LST. The result was similar to previous studies [49,50]. Among them, about 7.81% areas in the total showed significant ($p < 0.01$) and slight ($p < 0.05$) relations, and their spatial distribution (Figure 12b) was consistent with the areas in which LST exhibited a significant and slight increasing trend (Figure 9a), which proved that the development of the social economy could greatly promote the increase of LST.

The R-coefficient and its spatial distribution in the daytime and the nighttime were similar to that at annual scale (Figure 12). However, the areas with significant and slight relations in the daytime were distributed more than that in the nighttime, mainly due to socio-economic activities always being more active in the daytime, and the release of more energy than that in the nighttime. In addition, the relation between NTL and LST in the daytime could be further enhanced by solar radiation.

### 3.6. Effect of NDMI on LST Variation

The relation coefficient (R_coefficient) between the NDMI and LST ranged from $-0.86$ to $0.89$ for 2003$-$2021 (Figure 13a). About 15.07% of the region showed a significantly positive correlation (Figure 13b), mainly distributed in the region at the junction of Sanmenxia City, Luoyang City, and Nanyang city, the region along the border of northwest and south of the province with high elevation, while the significantly negative correlation was mainly distributed in urban and built-up areas. This phenomenon is similar to the correlation between NDVI and LST reported by Karnieli et al. and Raynolds et al. [21,51]. Generally, moisture presents a negative correlation with LST [22]. However, energy is the limiting factor of vegetation growth in high elevation areas, that is to say, the LST increase could promote vegetation growth, and the water supply for canopy increases correspondingly, so

the NDMI presented a positive correlation. The R-coefficient and its spatial distribution in the daytime and the nighttime were similar to that of the annual scale (Figure 13a–d), but the areas with a positive correlation shrunk in the daytime, while the areas with a negative correlation shrunk in the nighttime.

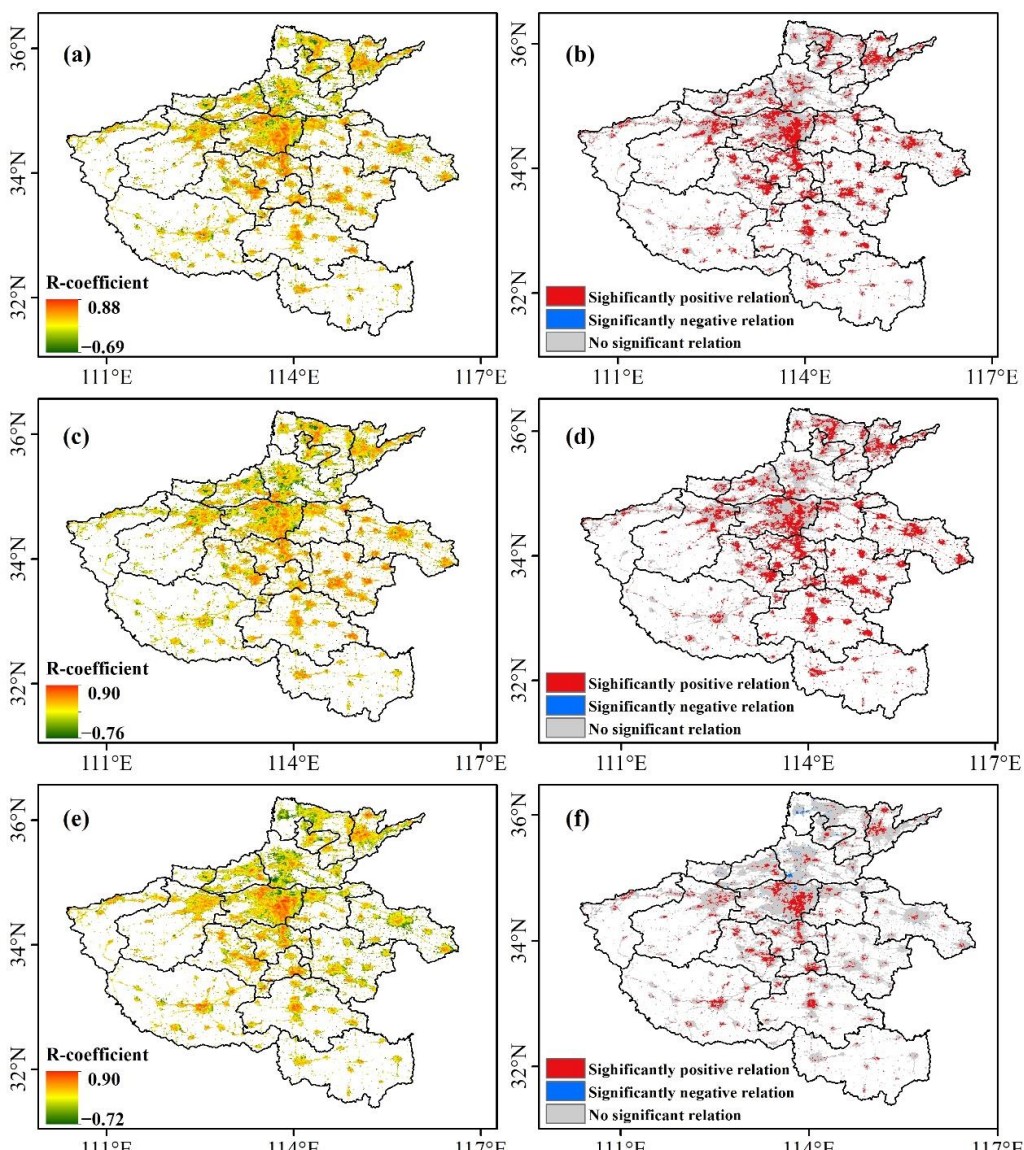

**Figure 12.** Relation coefficient and its significance between NTL and LST (white areas: no NTL) (**a**,**b**) interannual; (**c**,**d**) daytime; and (**e**,**f**) nighttime.

At seasonal scales, the spatial distribution of a significantly positive correlation in spring, summer, and fall was similar to that of the annual scale (Figure 14). Meanwhile, the spatial distribution of the significantly negative correlation presented an obvious difference. In spring, summer, and fall, the significantly negative correlation was mainly distributed in the cropland located in the eastern part of the province, but here, it presented a non-significant correlation at the annual scale. We speculate that the LST variation in cropland showed a discrepancy at different stages of the crop growth. The influence mechanism of the crop growth on the variation of LST in the cropland at different timescales needs to be further explored.

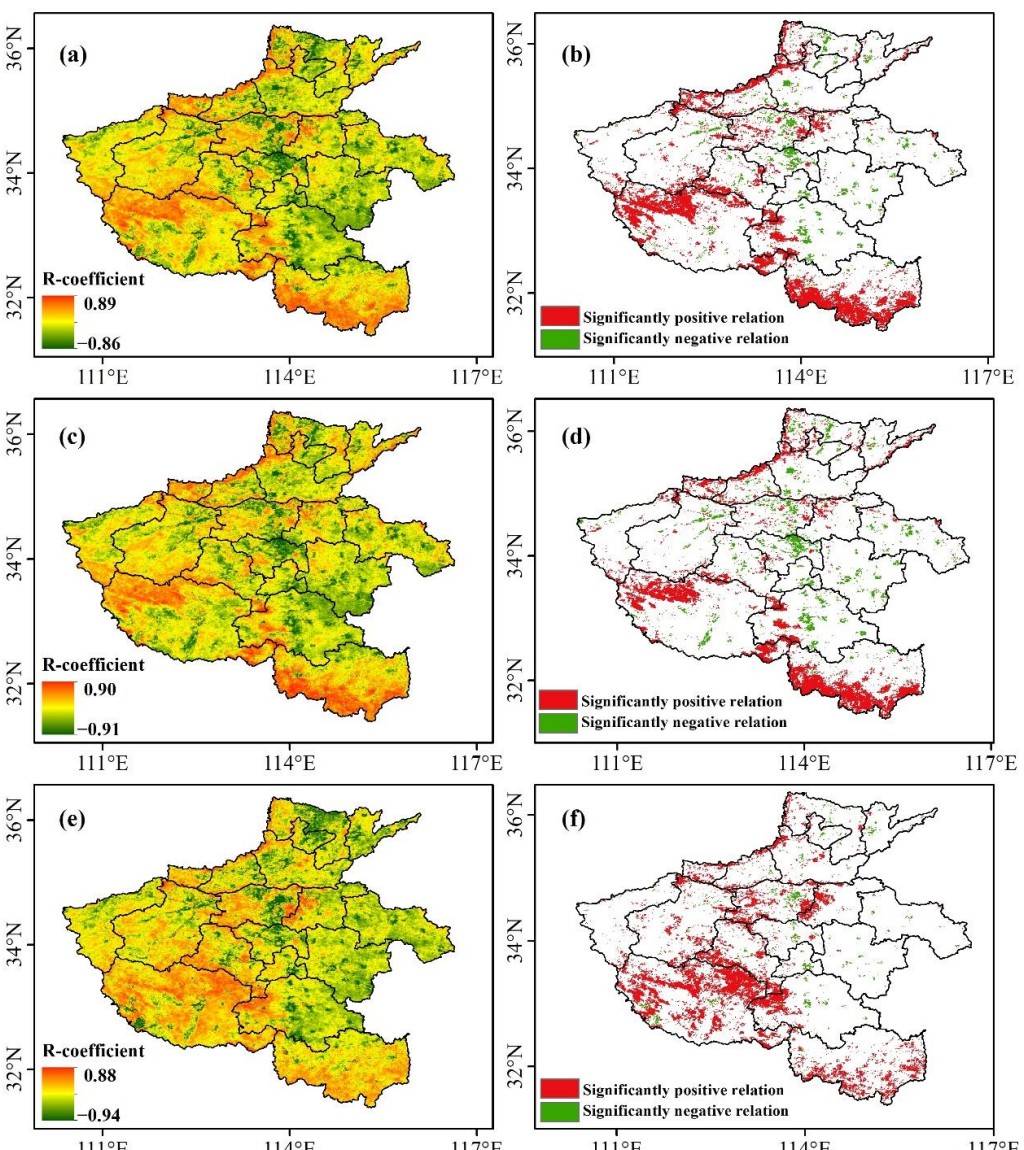

**Figure 13.** Relation coefficient and its significance between NDMI and LST (**a,b**) interannual; (**c,d**) daytime; and (**e,f**) nighttime.

The correlation coefficient between NDMI and LST of the daytime and the nighttime was further computed at seasonal timescales (Table 4). The absolute mean value of the correlation coefficient in the daytime higher than that in the nighttime, which indicated that the cooling effect of NDMI on LST in the daytime is greater than that in the nighttime, especially in summer.

**Table 4.** Correlation coefficient between NDMI and LST of daytime and nighttime at seasonal scale.

|  | Range | | Mean | |
|---|---|---|---|---|
|  | **Daytime** | **Nighttime** | **Daytime** | **Nighttime** |
| Spring | −0.98−0.82 | −0.88−0.86 | −0.39 | 0.04 |
| Summer | −0.97−0.83 | −0.91−0.88 | −0.46 | 0.19 |
| Autumn | −0.94−0.83 | −0.82−0.90 | −0.16 | 0.04 |
| Winter | −0.91−0.71 | −0.90−0.86 | −0.24 | −0.06 |

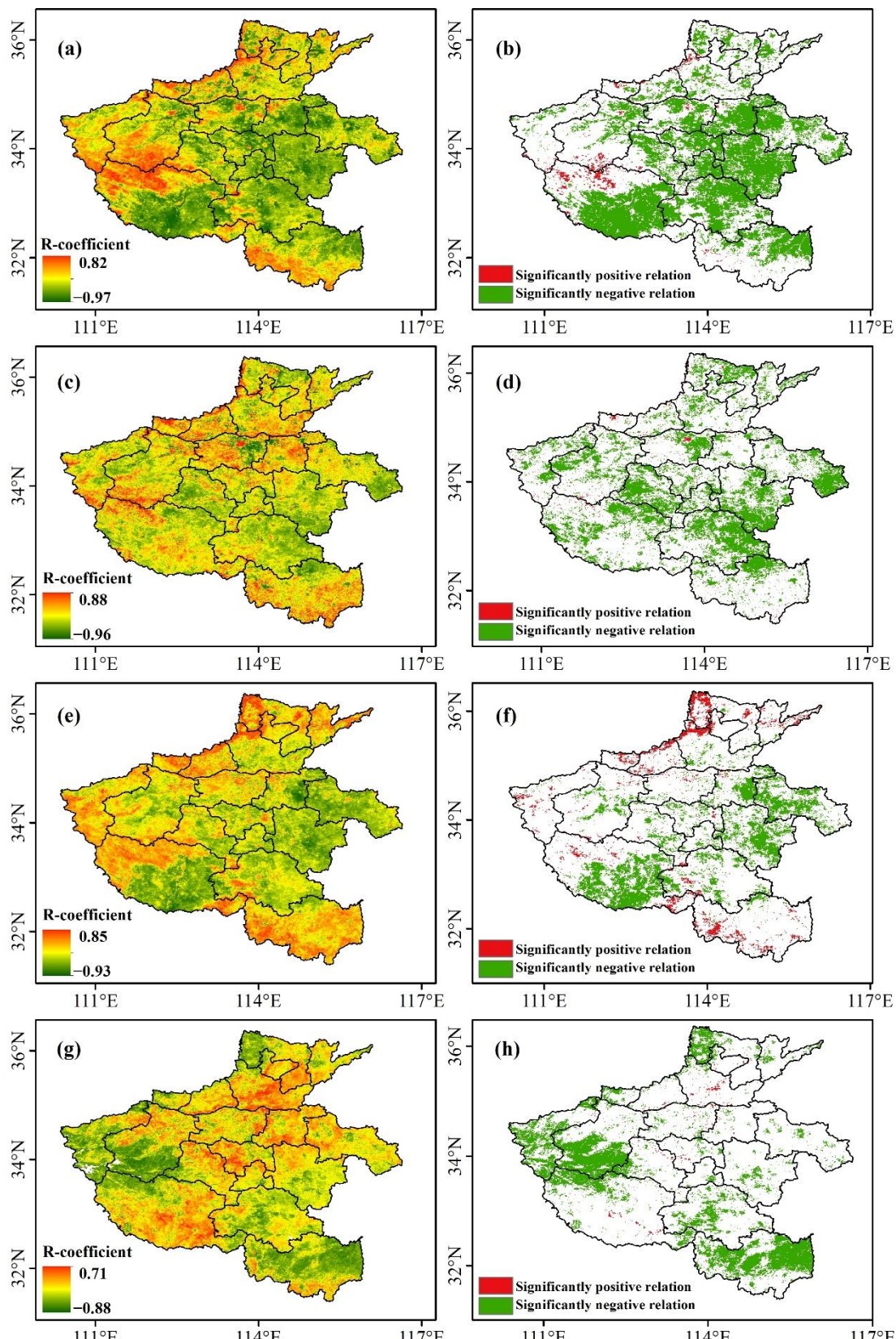

**Figure 14.** Relation coefficient and its significance between NDMI and LST at seasonal scales (**a**,**b**) spring; (**c**,**d**) summer; (**e**,**f**) fall; and (**g**,**h**) winter.

## 4. Conclusions

Henan Province of China experienced rapid development since "the Rise of Central China" strategy was put forward, and this exerted an influence on the eco-climate. In this study, the spatiotemporal pattern of LST and its trend characteristics in Henan Province were investigated based on MODIS LST from 2003 to 2021. In addition, the effect of land cover types, Nighttime Light data, and NDMI on LST were discussed. The main conclusions were summarized as follows:

(1)  The LST experienced the steady and rapid decreasing for 2004–2010 and 2018–2020, respectively, whereas an obvious increase and slight increase occurred for 2010–2013 and 2014–2018, respectively.

(2)  The spatial pattern presented a high temperature in the central part, while presenting a low temperature in the western part of the province at the annual and daytime scales. In the nighttime, the spatial distribution of the LST exhibited decreased from the southern part to the northern part of the province. At the seasonal scales, the spatial pattern of LST in spring and summer was similar to that in the daytime, while the pattern in fall and winter was basically consistent with that of the nighttime. The largest Standard Deviation (STD) was observed in summer, which indicated the highest spatial variation of the province in summer.

(3)  The LST exhibited a warming trend; the interannual variation rate of LST was 0.08 °C/Y. An increasing trend mainly occurred in the urban and built-up areas. At the seasonal scales, the rate decreased sequentially in the order of fall, winter, spring, and summer. In addition, the LST difference between the daytime and the nighttime gradually increased, especially in fall, due to the rising rate in the daytime being higher than that in the nighttime.

(4)  The LST increase along the expansion of the urban and built-up lands and socio-economic development on the whole. The correlation between LST and NDMI showed a significant difference at the spatiotemporal scale. At the annual scale, NDMI in the western part with high elevation presented a significantly positive correlation with LST, and a significantly negative correlation in urban and built-up areas, whereas a significantly negative correlation mainly occurred in the cropland located in the eastern part during crop growth in spring, summer, and fall. The cooling effect of NDMI on LST in the daytime was greater than that in the nighttime.

The limitations should be pointed out that this paper only examined the variation of LST under clear-sky conditions, and the influence of cloud on LST variation in Henan Province was ignored. In addition, we found that, in croplands located in the eastern part of the province, NDMI showed a non-significant correlation with LST at annual scale, and a significantly negative correlation with LST in spring, summer, and fall. Therefore, the influence mechanism of cropland on the variation of LST at different timescales needs to be further explored.

**Author Contributions:** Conceptualization, Z.Q. and S.L. (Shifeng Li); methodology, S.L. (Shilei Li); validation, S.L. (Shifeng Li), Q.L. and S.L. (Shilei Li); formal analysis, S.L. (Shifeng Li); writing—original draft preparation, S.L. (Shifeng Li); writing—review and editing, S.Z., M.G. and W.D.; supervision, W.D.; project administration, Z.Q.; funding acquisition, Z.Q. All authors have read and agreed to the published version of the manuscript.

**Funding:** This study was supported by the National Key Research and Development Program of China (Grant No.: 2019YFE0127600, 2016YFA0600302) and the National Natural Science Foundation of China (Grant No.: 41771406, 41921001).

**Institutional Review Board Statement:** Not appliable.

**Informed Consent Statement:** Not appliable.

**Data Availability Statement:** The data that support the findings of this study are available from the first author upon reasonable request.

**Conflicts of Interest:** The authors declare no conflict of interest.

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
