# Peer review of "Spatiotemporal Variation of Land Surface Temperature in Henan Province of China from 2003 to 2021"

_land, doi:10.3390/land11071104_

Round 1

Reviewer 1 Report

The authors seem to have addressed my main concerns in the last round of review. However, seen from their response letter and the revised parts, the way that the authors responded and solved the issues is not in a rigorous sense. They 'strategically' used 'to-focus-on' and evaded the crucial point, and some of my specific suggestions (like the key related advances studies) were not got answered facedly even. The depth of the background knowledge/discussion and precision of the relevant literature review, I did suggest, can be improved to guarantee the fundamentals of this study are solid and clear, and the communication with the current global active peers is sufficient. 

Personally, I liked the paper basically, from its topic and the solid content, and very believed it has the potential to be published. (That is why I spent such much time commenting here.) By this round of feedback and revision, however, at present, my devotion and suggestions seemed not to receive equal echos. Please do not rush and carefully perfect the manuscript. 

Here are the authors' responses passed by the editor. I read it and added further comments in this round.

Point 1: At the beginning of the Introduction, I expected to see the authors differentiate the global climate system and the regional climate system clearly but the present statement seems to have confused them together.

Response 1: Thanks for the suggestion. We have revised the expression in the new revision. In new revision, we mainly focus on the importance of LST, and no longer talk about global climate change. After modification, the expression seems clearly and conforms to the subject. 

[New question:---what they (regional land surface climate system and global climate system) are different and how they are connected are what I really want to know from the perspectives / logical summary provided by the authors. I did not mean that let you simply cut and threw away the other minor but relevant aspect.]

[examples of 'how to draw a conceptualized diagram':

(2020). Urbanization-driven changes in land-climate dynamics: a case study of Haihe River Basin, China. Remote Sensing

(2021). Assessing spatiotemporal characteristics of urban heat islands from the perspective of an urban expansion and green infrastructure. Sustainable Cities and Society

Point 2: Another main problem that needs to be addressed in the revision is too many descriptive statements.

Response 2: Thanks for the suggestion. We have revised some of the expression in the new revision. In new revision, we are not only describe the results, but also add some conclusive expressions. 

[New question:---Adding more conclusive expressions is not enough. My key point meant that you should present a coherent, valuable, logically organized line to explicate the major novel findings.]

Point 3:  For my suggestions, first to make the results more concise and concentrated and avoid stating repetitive information, second to refer the updated research in the field of land-climate interaction, mainly focused on the objects of NDVI and land surface heat environment, and its implications, adding some theoretical summary diagram to conceptualize the key findings or the mechanism principles.

Response 3: Thanks for the suggestions. In new revision, we mainly focus on spatiotemporal variation of LST at different timescales in Henan Province, and analyze the influences of land cover types, NDMI (Normalized Difference Moisture Index), and NTL (Nighttime Light) data. Therefore, we don’t analyze in detail the impact of LST changes on the land-climate interaction. 

[New question:---For this answer, it seems that the authors totally overlooked my suggestion to add a conceptualized diagram which it would make your study more systematically organized and deliver higher value to the theory of eco-climatology as a piece of empirical evidence. Please carefully rethink the logic in our discussion here: analyzing the influences of land cover types, NDMI, and NTL data, integrated with the LST, belongs to the field of the land-climate interaction. I had not meant to suggest you add a more detailed analysis of the impact of LST on the land-climate interaction. So, why you said that 'we don't ....'.]

=============================

Here are my complete comments in the previous round of review for your referring.

In this manuscript, the authors presented a study about the relationship of the spatial-temporal pattern between the vegetation and the land surface temperature applying satellite imagery datasets in a case of Henan Province in China, which fitted the journal theme and the Section of 'Land–Climate Interactions'. However, the writing needs to be well improved before getting published on Land. I have two major concerns and suggestions on the current version as the following. 

1. The background knowledge is not solid and deep adequately. At the beginning of the Introduction, I expected to see the authors differentiate the global climate system and the regional climate system clearly but the present statement seems to have confused them together. Global warming or the greenhouse effect is from the perspective of the earth system at the continental scale, while land surface temperature dynamics means the underlying surface heat environment at the regional scale which can be observed and measured via remote sensing approaches and be closely interacted with the vegetation. I suggested reading and absorbing more theories in eco-climatology, written by Gordon B. Bonan for instance. The adaptation and mitigation for climate change could be connected to the land surface temperature dynamics, but the logic between them should be clearly constructed for better justifying or interpreting your research in the first and the by the end.

2. Another main problem that needs to be addressed in the revision is too many descriptive statements. It made the whole manuscript read like an experimental or data report rather than scientific research to bring more useful insights and novel discoveries or at least testification on some theories to the potential readers. It is easier to demonstrate the facts from spatial statistics but too many similar studies have been published and documented and such a descriptive manner become a common commercialized way in the industries. For an academic paper, we are looking for a more original and methodological novelty to be developed. For my suggestions, first to make the results more concise and concentrated and avoid stating repetitive information, second to refer the updated research in the field of land-climate interaction, mainly focused on the objects of NDVI and land surface heat environment, and its implications, adding some theoretical summary diagram to conceptualize the key findings or the mechanism principles. For references, around my hands, I suggested several relevant active world-class scholars/groups could be followed and learned: prof. Xuhui Lee from Yale University, dr. Lei Zhao from University of Illinois, Urbana-Champaign, (associate) prof. Yan Li from Beijing Normal University, (associate) prof. Adriaan J. Teuling from Wageningen University and Research.

Author Response

Point 1: [New question:---what they (regional land surface climate system and global climate system) are different and how they are connected are what I really want to know from the perspectives / logical summary provided by the authors. I did not mean that let you simply cut and threw away the other minor but relevant aspect.]

Response 1: Thanks for the constructive suggestion. After receiving the question, I downloaded some papers including the papers you suggested on this question. In my view, regional climate change drives and amplifies global climate change, and regional land surface climate system is influenced directly by human activities, such as urbanization, industrialization and so on. As a parameter of regional land surface climate system, LST varies mainly due to human activities, not global warming. Therefore, according to the subject in my research, the introduction beginning will be rephrased inn new manuscript as follows:

Global warming is one of dramatic challenges currently faced by international community [1-2]. In the context of global warming, the effects of human activities on climate change have been concerned by much more scholars [3-5]. Along with the rapid urbanization and industrialization, the land surface properties are altered significantly and the regional hydrothermal environment is reshaped [6-8]. As a key parameter closely related to various land surface processes and energy balance, long-term LST can be an effective indicator of surface-atmosphere interactions. Therefore, it is necessary to analyze a long-term spatiotemporal variation of LST to understand the characteristics of land-climate.

Point 2: [New question:---Adding more conclusive expressions is not enough. My key point meant that you should present a coherent, valuable, logically organized line to explicate the major novel findings.]

Response 2: Thanks for the suggestion. We have added conceptualized diagram in the new manuscript.

Point 3: [New question:---For this answer, it seems that the authors totally overlooked my suggestion to add a conceptualized diagram which it would make your study more systematically organized and deliver higher value to the theory of eco-climatology as a piece of empirical evidence. Please carefully rethink the logic in our discussion here: analyzing the influences of land cover types, NDMI, and NTL data, integrated with the LST, belongs to the field of the land-climate interaction. I had not meant to suggest you add a more detailed analysis of the impact of LST on the land-climate interaction. So, why you said that 'we don't ....'.]

Response 3: Thanks for the insightful suggestions. I am sorry to overlook the suggestion to add a conceptualized diagram in the last review. In the new manuscript, we have added conceptualized diagram.

In addition, NDMI is an indicator of land surface moisture, and moisture is a parameter of climate. In fact, it is similar to NDVI but has closer relation to LST than NDVI. NTL data can represent socio-economic development and people agglomeration at some extent, which is similar urban heat island effect. Therefore, analyzing the influences of NDMI is related to the land-climate interaction directly or indirectly.

Reviewer 2 Report

This manuscript is aiming in studying the spatiotemporal variations of land

surface temperature over the Henan Province in China for a 19-year period

(2003-2021) and in extracting information regarding the eco-climatic

characteristics of the area. Additionally, the authors wanted to investigate the

effect of land cover types, Nighttime Light data (NTL) and Normalized

Difference Moisture Index (NDMI) on LST.

The main objectives of the study are covered in a way that could use some

improvement. In general the text that I have received is in the track-changes

mode which should be changed and a more clear document can be sent.

Also English language can also be improved. In addition, the figures and

tables should be placed in the text so each figure/table to be seen as a

whole.

Comments/suggestions and questions:

- Maps with absolute values of the Nighttime Light data and Normalized

Difference Moisture Index used for the comparison with MODIS LST and the

analysis performed should be shown in the manuscript as a reference for

the reader.

- Conclusions can be written in a more extensive and slightly more detailed

way to better address the results of the study.

- line 49: Currently, a great deal of research focus

Currently, a great deal of research focuses

- line 55: “... the spatiotemporal variation of temperature due to exhibit …”

“… the spatiotemporal variation of temperature due to exhibiting …”

- line 57: “..has been a unique data..”

“..has been a unique dataset..”

- line 64: “...LST for different region.”

“..LST for different regions.”

- line 71: “... ACP model [20]”

Need to define ACP

- line 78: “... folks mainly focus ..”

Please substitute the word “folks” with another.

- Line 80-82: “But the influence of NDMI on spatiotemporal variation of time-

series LST was seldom to investigated.”

Please rephrase this sentence: “But the influence of NDMI on spatiotemporal

variation of time-series LST was rarely investigated.”

- line 83: “...to lack the data with spatial continuity. “

“ .. to lack of data with spatial continuity.”

line 85: “...of carbon omission ..”

Are you referring to omission or emission?

- Line 89: “Henan Province of China have experienced …”

“Henan Province of China has experienced …”

- line 151: “...data accessibility was choose…”

“...data accessibility was chosen…”

- line 197-198: Why is i=1-19? Is this representative to the time period that

you are studying? (2003-2021)

- line 249: “It indicating that the variation of LST ..”

“It indicated that the variation of LST ..”

- line 285: “... it indicating the highest spatial variation…”

“... it indicated the highest spatial variation…”

- lines 331-332: “The LST difference among different land cover was similar

in annual, spring, summer, and fall, but different from winter.”

Please rephrase! You are using too many times the word “different”.

- lines 378-382: “However, the significant and slight relation in daytime were

distributed more than that in nighttime, it indicated that socio-economic

activities played a more obvious role in promoting the increase of LST in

daytime than that in nighttimedue to socio-economic activities were more

active in daytime, and released more energy than that in nighttime.”

This is a huge sentence. Please rewrite it into smaller.

The authors use the expression “...it indicating…” many times in the text.

Please consider minimizing the times this phrase is used and substituting it

with a more relevant to the context.

Author Response

Point 1: Maps with absolute values of the Nighttime Light data and Normalized Difference Moisture Index used for the comparison with MODIS LST and the analysis performed should be shown in the manuscript as a reference for the reader.

Response 1: Thanks for the suggestion. In new manuscript, we have added two maps about the Nighttime Light data and Normalized Difference Moisture Index in data sources section, and make the analysis related.

Point 2: Conclusions can be written in a more extensive and slightly more detailed way to better address the results of the study.

Response 2: Thanks for the suggestion. We have rephrased the conclusion in the original basis as presented in new revision. In new revision, four points are listed, including temporal fluctuation, spatial variability, trend characteristics, and drive mechanism of land cover types, NTL, and NDMI at different timescales.

Point 3: line 49: Currently, a great deal of research focus

Currently, a great deal of research focuses

- line 55: “... the spatiotemporal variation of temperature due to exhibit …”

“… the spatiotemporal variation of temperature due to exhibiting …

line 57: “..has been a unique data..”

“..has been a unique dataset..”

- line 64: “...LST for different region.”

“..LST for different regions.”

- line 71: “... ACP model [20]”

Need to define ACP

- line 78: “... folks mainly focus ..”

Please substitute the word “folks” with another.

- Line 80-82: “But the influence of NDMI on spatiotemporal variation of time-

series LST was seldom to investigated.”

Please rephrase this sentence: “But the influence of NDMI on spatiotemporal

variation of time-series LST was rarely investigated.”

- line 83: “...to lack the data with spatial continuity. “

“ .. to lack of data with spatial continuity.”

line 85: “...of carbon omission ..”

Are you referring to omission or emission?

- Line 89: “Henan Province of China have experienced …”

“Henan Province of China has experienced …”

- line 151: “...data accessibility was choose…”

“...data accessibility was chosen…”

- line 197-198: Why is i=1-19? Is this representative to the time period that

you are studying? (2003-2021)

- line 249: “It indicating that the variation of LST ..”

“It indicated that the variation of LST ..”

- line 285: “... it indicating the highest spatial variation…”

“... it indicated the highest spatial variation…”

- lines 331-332: “The LST difference among different land cover was similar

in annual, spring, summer, and fall, but different from winter.”

Please rephrase! You are using too many times the word “different”.

- lines 378-382: “However, the significant and slight relation in daytime were

distributed more than that in nighttime, it indicated that socio-economic

activities played a more obvious role in promoting the increase of LST in

daytime than that in nighttime due to socio-economic activities were more

active in daytime, and released more energy than that in nighttime.”

This is a huge sentence. Please rewrite it into smaller.

The authors use the expression “...it indicating…” many times in the text.

Please consider minimizing the times this phrase is used and substituting it

with a more relevant to the context.

Response 3: Thanks for the suggestions. We have revised the manuscript as suggested.

This manuscript is a resubmission of an earlier submission. The following is a list of the peer review reports and author responses from that submission.